# The Spectrum of Cognitive Dysfunction in Amyotrophic Lateral Sclerosis: An Update

**DOI:** 10.3390/ijms241914647

**Published:** 2023-09-27

**Authors:** Kurt A. Jellinger

**Affiliations:** Institute of Clinical Neurobiology, Alberichgasse 5/13, A-1150 Vienna, Austria; kurt.jellinger@univie.ac.at; Tel./Fax: +43-1-5266534

**Keywords:** ALS, cognitive impairment, frontotemporal lobe degeneration, multimodal neuroimaging, brain hypometabolism, functional connectivity, cognitive reserve

## Abstract

Cognitive dysfunction is an important non-motor symptom in amyotrophic lateral sclerosis (ALS) that has a negative impact on survival and caregiver burden. It shows a wide spectrum ranging from subjective cognitive decline to frontotemporal dementia (FTD) and covers various cognitive domains, mainly executive/attention, language and verbal memory deficits. The frequency of cognitive impairment across the different ALS phenotypes ranges from 30% to 75%, with up to 45% fulfilling the criteria of FTD. Significant genetic, clinical, and pathological heterogeneity reflects deficits in various cognitive domains. Modern neuroimaging studies revealed frontotemporal degeneration and widespread involvement of limbic and white matter systems, with hypometabolism of the relevant areas. Morphological substrates are frontotemporal and hippocampal atrophy with synaptic loss, associated with TDP-43 and other co-pathologies, including tau deposition. Widespread functional disruptions of motor and extramotor networks, as well as of frontoparietal, frontostriatal and other connectivities, are markers for cognitive deficits in ALS. Cognitive reserve may moderate the effect of brain damage but is not protective against cognitive decline. The natural history of cognitive dysfunction in ALS and its relationship to FTD are not fully understood, although there is an overlap between the ALS variants and ALS-related frontotemporal syndromes, suggesting a differential vulnerability of motor and non-motor networks. An assessment of risks or the early detection of brain connectivity signatures before structural changes may be helpful in investigating the pathophysiological mechanisms of cognitive impairment in ALS, which might even serve as novel targets for effective disease-modifying therapies.

## 1. Introduction

Amyotrophic lateral sclerosis (ALS) is a fatal multi-system neurodegenerative disease with no effective treatment or cure. It is characterized primarily by motor neuron degeneration but may be accompanied by cognitive dysfunction. The involvement of respiratory muscles is a key feature and usually leads to death due to respiratory failure. The mean survival time is 2–5 years after disease onset [1]. Classification of the disease is based on the site of origin (spinal or bulbar onset), the severity of involvement, and the pattern of heritability. ALS has an incidence between 1.5 and 2.5 per 100,000/year and a point prevalence of around 5 per 100,000 population, with local variations [2,3,4,5]. ALS, which has an estimated heritability between 40 and 50% [6], occurs sporadically and familial, with an autosomal-dominant inheritance (5–10%) [7]. Four genes are associated with up to 70% of familial ALS (fALS), namely C9orf72 (chromosome 9 open reading frame 72), TARDBP (transactive response DNA binding protein), SOD1 (superoxide dismutase 1), and FUS (fused-in sarcoma protein) [8], but there is a long and growing list of mutations found in sporadic ALS (sALS) and/or fALS. The frequency of C9orf72 mutation is low in Japan, unlike in Europe and the USA, while SOD1 and FUS are more common [9]. ALS heritability is enriched in binding sites of RNA-binding proteins, including TDP-43 and FUS [10]. Genetic polymorphisms may exert a polygenic contribution to the risk of cortical disease and cognitive dysfunction in ALS [11]. Neuropsychiatric disorders in parents and the families of ALS patients are associated with cognitive (and behavioral) changes, such as poorer overall cognition and visuospatial or language disorders, overlapping with ALS or other motor neuron diseases [12].

At the onset, signs and symptoms of ALS are rather subtle, with the progressive involvement of the lower and upper motor neurons. The site of symptoms is important because of distinguishing between the limb, bulbar and respiratory onset of different ALS subtypes. Clinical diagnosis is usually made according to the revised El Escorial criteria [13], further modifications of which are aimed at increasing diagnostic sensitivity [14]. The three major ALS variants are sALS, fALS (about 20%), and the rare Western Pacific form. There is an overlap between ALS and frontotemporal lobe degeneration (FTLD), with the latter presenting with cognitive impairment (CI) progressing to dementia. The ALS-FTLD phenotype with no to severe CI is accompanied by ALS pathology and its clinical presentation. The most consistent ALS and ALS-FTLD pathology is a disturbance in transactive response DNA-binding protein 43 kDA (TDP-43) metabolism, observed in up to 97% of all cases [15,16], with phosphorylated TDP-43 pathology causing a disruption of synaptic structures [17,18]. Alterations in tau protein metabolism have also been observed in ALS-FTLD [19]. During the last 10 years, radiological, humoral, neurophysiological and genetic biomarkers have been reported in ALS and some of them correlate to the CI of ALS patients. Early detection and careful monitoring of ALSci even at early disease stages are crucial for patient and caregiver and may not only have prognostic implications but may also be important for future clinical trials [20]. The present review is restricted to the neuroimaging and pathological bases of CI in ALS and its pathogenesis, without a specific discussion of FTLD.

## 2. Cognitive Abnormalities in ALS

Cognitive (and behavioral) impairment has been known since the 19th century thanks to Biblioteca di Area Medica “Adolfo Ferrate” Sistema Bibliotecario di Ateneo, University of Pavia, Italy [21]. CI in ALS is more frequent than in patients with other neuromuscular diseases [22], and in the last decade, it has become clear that cognitive (and behavioral) changes in ALS are more frequent than previously recognized [23]. The incidence of CI in ALS varies considerably between 30% [22,24,25,26,27,28,29,30,31] and 75% [19,32,33], whilst the rate of behavioral dysfunction occurs in up to 50% of ALS patients [32]. The published data about the severity of CI in ALS vary considerably: mild cognitive impairment (MCI) was observed between 32% and 70% [33,34], and between 6% and 40–45% showed severe CI [33,34], while others reported frank dementia in 15% of ALS patients [19] with full symptoms of FTLD, predominantly the behavioral variant [35]. Cognitive functions were more impaired in patients with an age of onset over 65 years [33]. Moreover, 14% of ALS patients had evidence of CI without executive dysfunction, and no cognitive abnormality was detected in 24% to 46.9% [25,29,35].

In a recent study from China, the ALSci group (49.5% of a total of 1015 ALS patients) tended to harbor more ALS-plus (extramotor) symptoms than the cognitively normal ALS group [36]. In non-demented ALS patients, no cognitive/behavioral correlate for hippocampal dysfunction was found [37].

The prevalence and pattern of CI in fALS is similar to that in sALS. For patients with fALS, the site of symptoms does not correlate with CI, but age does [38]. The cognitive spectrum in ALS is heterogenous and consists of deficits in executive function (EF), attention, verbal fluency, naming, language, social cognition, visuospatial abilities, verbal memory and other cognitive domains [24,30,39,40,41,42,43]. Foremost, deficits are observed in the domains of attention, executive function and language function [35,44]. Memory deficits are distinctly different from those in amnestic MCI, which may be due to the fact that they are not exclusively caused by coexisting executive dysfunction in ALS. This indicates qualitative differences in the type of memory impairment between ALS and MCI and supports that temporal dysfunction is related with distinct cognitive impairments in ALS [45].

There is emerging evidence that presymptomatic disease is not uniformly clinically silent with mild motor, cognitive and/or behavioral impairment, representing a prodromal stage of ALS [20]. In contrast to motor functioning that declines concomitantly with disease progression, cognitive deficits appear in early disease, are essentially present at the initial visit and usually do not substantially decline on follow-up [46]. Cognitive deterioration in non-demented patients with ALS is a relatively slow process; selected CI in the form of verbal fluency deficits appears relatively early in the course of the disease, although language function may become vulnerable as the disease progresses [47]. Despite vast evidence for CI at disease onset in different domains, evidence for the evolution of these deficits, however, is rather limited, suggesting that ALS patients present with CI early in the course of disease [48]. Language impairment is characteristic of ALS in early disease stages and can develop independently of executive dysfunction, reflecting selective patterns of frontotemporal involvement [49]. Occurring in about 23% of non-demented ALS patients, it may also be significantly driven by executive dysfunction, but independent of other motor and non-motor features [50]. The spectrum of language impairment in ALS includes spontaneous speech, word fluency and action naming, whereas object naming, semantic memory, sentence comprehension and repetition are comparable to healthy controls. Spontaneous speech is reduced across the ALS spectrum even in those with intact core language abilities. Around 36% of ALS patients produced significantly fewer words per minute, suggesting that spontaneous speech is reduced across the ALS spectrum [26]. The frequency of language abnormalities in ALS was 8.5% for spontaneous speech, 25.7% for verbal comprehension, 8.8% for repetition, 14.7% for naming, 17.6% for reading, and 51.4% for writing, with dysfunction on the left angular gyrus probably being associated with writing errors observed in ALS [51]. A recent study reported a prevalence of language impairment of 22.7%, which is significantly driven by executive dysfunction and not associated with other motor or non-motor features [50].

While executive dysfunctions have been described first, alterations in verbal fluency, pragmatic and social cognition have also been observed [52]. The dysfunction of social cognition is well recognized as one of the CIs of ALS [42]. This is a new domain with a relatively large effect size, highlighting the overlap between ALS and FTLD [39]. ALS-FTLD patients have a worse prognosis and shorter survival rates than patients with pure ALS [53].

CI in both fALS and sALS may present before, at, or after the onset of motor neuron disease [25,54]. It usually develops without bias to movement impairment [39,55,56], whereas others have stated that cognitive deficits (and behavioral impairment) accompany the intensifying physical activity and are more frequent with a more severe disease stage [57]. In ALS, MCI is associated with exaggerated gait variability and an increased risk of falls [58]. A high level of CI arrives within 18 months of symptom onset [59], while others reported the worsening of CI after 6 months [25]. The decline in cognitive function was faster in patients who were cognitively impaired at baseline, while normal cognition at baseline was associated with a tendency to remain cognitively intact, with slowed motor and cognitive progression [60]. Weight loss, very common in ALS, is associated with executive dysfunction and cognitive decline, as well as poor survival [61], whereas there is no significant difference in the survival curves of pure ALS and ALS with cognitive and behavioral changes [62]. It was suggested that baseline executive status might predict survival in ALS [63].

Patients with bulbar onset and those with bilateral spinal onset were generally characterized by lower cognitive performance in most neuropsychological tests when compared to patients with lateralized onset, which was in keeping with the hemispheric lateralization of language and visuospatial abilities [64]. Cognitive deficits are more pronounced in ALS with predominant upper motor neuron (PUMN) involvement [19,65]. Some authors stated that cognitive changes are more frequent with more severe disease stages [57], whereas, according to others, cognitive performance remains stable [48]. ALS patients with dysarthria performed significantly worse than non-dysarthric patients, while cognitive functioning was not associated with the site of onset and had only a moderate association with dysarthria [66]. Others also stated that abnormalities in cognitive domains have only limited relevance for the patient’s everyday life, in comparison to the impact of behavioral alterations [67].

C9orf72 repeat expansions seem to cause a recognizable phenotype characterized by earlier disease onset, more severe CI, more rapid progression and shorter survival [68,69,70,71,72,73]; neuropsychological changes of these patients are associated with subcortical gray matter (GM) atrophy [74]. Likewise, those with dysarthria, family history of ALS, PUMN phenotype and bulbar onset have a higher risk for ALSci [75]. By contrast, patients with the atypical ALS type 8, caused by a mutation of the vesicle-trafficking protein VAPB [76], demonstrated similar deficits in most cognitive domains as patients with sALS [77], while the landscape of CI in SOD1 ALS is under discussion [78].

## 3. Neuroimaging Findings in ALSci

Early MRI and SPECT studies showed frontal atrophy, suggesting a continuum of cognitive disability in ALS patients corresponding to a pathological process in the frontal lobe ranging from normality to significant impairment [79]. Later studies suggested that cognitive dysfunction in ALS reflects functional and morphological deficits outside the primary motor system that are specific to the nature and evolution of the basic disease [46] (Table 1). While structural neuroimaging has not been able to establish a specific hypothesis of extramotor cortical atrophy beyond the combination of various frontal, temporal and limbic areas, functional neuroimaging supports the neuropsychological findings of frontotemporal dysfunction but also implies the involvement of somatosensory areas [80]. A significant decline in the cortical thickness of frontal, temporal and parietal regions is observed during disease progression, whereas the reduced cortical thickness of the precentral gyrus at the beginning of the disease remains stable [81,82]. Multimodal neuroimaging confirmed motor and non-motor GM and WM abnormalities in non-demented cognitively impaired ALS patients, as well as early extramotor pathologies in those without CI [83]. Cortical thinning in the frontoparietal region suggesting a disease-specific pattern of neurodegeneration was present in all ALS patients, irrespective of cognitive (and behavioral) status, whereas the thinning of inferior frontal, temporal, cingulate and insular cortices was typical for cognitive and behavior deficits, with language impairment mainly related to the left temporal pole and insular involvement [84] (Table 1). Menke et al. [85] assessed brain changes in patients with ALS over 2 years and reported widespread changes in both WM and GM in the cingulate gyrus, thalamic nuclei, caudate nuclei, pallidum, hippocampi, parahippocampal gyri, and insula (Table 1). These results indicate that over time, cerebral changes extend into extramotor areas, but it remains difficult to draw a link between these changes and clinical cognitive implications [86]. Cognitive changes are highly dependent upon cortical atrophy; patients with ALSci have significant atrophy across motor and somatosensory as well as adjacent frontal and parietal areas [87]. Subcortical GM structures are often involved before CI becomes evident [88]. The degeneration of basal ganglia in ALS is particularly seen in patients with CI (and behavioral involvement); thalamic atrophy is associated with poorer EF, and reduced volumes of globus pallidus, caudate nucleus, putamen and thalamus are associated with poorer cognitive scores [89]. A significant reduction in amygdala volume may also play an important role in cognitive dysfunction in ALS [90]. Parts of the hypothalamus, containing the paraventricular nucleus, show essential volume loss in ALS and ALS + FTD [91].

While ALS-motor patients showed decreased GM volume in frontal and cerebellar areas and increased GM volume in the right supplementary motor cortex, ALSci patients showed diffused GM volume reduction in the primary motor cortex bilaterally, and frontotemporal, bilateral cerebellum, basal ganglia and bilateral superior longitudinal fasciculi white matter (WM) were seen [97]. The cortical thickness of the left entorhinal cortex, left inferior temporal gyrus, left medial orbitofrontal lobe and left insular lobe were significantly reduced (*p* < 0.05). This correlated with neuropsychological scores [31]. There was also decreased fractional anisotropy (FA) and increased radial diffusivity in the corticospinal tract bilaterally, corpus callosum, and extramotor tracts in ALS-motor patients, and decreased FA and increased axial diffusivity in motor and several WM tracts in ALSci patients [111].

C9orf72 ALS mutation carriers exhibited extensive cortical and subcortical damage with disease-specific patterns of thalamo-cortico-striatal atrophy. Cortical thinning was pronounced in posterior areas and extended to frontal regions. Bilateral atrophy of the mediodorsal and pulvinar nuclei emphasized a focal rather than global thalamus atrophy, indicating a central role of the thalamus in the pathogenic mechanisms of C9orf72-mediated disease [112]. C9orf72-repeat-negative ALS patients showed a volume reduction in the right nucleus accumbens, left caudate, and hippocampus [103].

Verbal fluency was associated with temporal lobe dysfunction, with increasing frontal involvement in ALS patients with greater CI [113]. Dissociated motor and cognitive components of speech deficits were related to cortical thickness in the primary motor cortex and perisylvian regions, and impaired speech duration was linked to the bilateral frontal operculum and left anterior insula [94]. Multifocal deficits in sentence expression that are independent of their motor disorder were related to GM atrophy in the left inferior frontal and anterior temporal regions and to reduced FA in superior longitudinal and inferior frontal-occipital fasciculi [93]. ALSci patients demonstrated cortical thinning in the bilateral precentral gyrus, right precuneus, and right frontal and temporal lobes; considerable frontoparietal atrophy, including the right insula and motor cortices, was associated with the involvement of the bilateral pallidum and putamen, with “pure” ALS displaying greater parietal atrophy [89]. These and other data support a more integral framework for the clinical relationship of frontal functioning as a correlate of cognitive deficits and executive dysfunction [114]. They furthermore support the concept that ALS patients with CI have a more widespread neurodegenerative process compared with a pure motor disease [107].

Resting-state functional MRI studies showed significant differences in the fractional amplitude of low-frequency fluctuation (fALFF)—(The amplitude of low-frequency fluctuation (ALFF) measures low-frequency oscillations of the blood-oxygen-level-dependent (BOLD) signal, characterizing local spontaneous activity during the resting state. ALFF is a commonly used measure for resting-state functional magnetic resonance imaging (rs-fMRI) in numerous basic and clinical neuroscience studies. (Verbatim from [115] in the left frontal lobe, right parietal lobe, and left cingulate gyrus, and there was regional homogeneity (ReHo) in these areas compared to those with and without CI. These changes were mainly located in the left prefrontal lobe and anterior cingulate cortex, indicating that hypoactivities are detected in extramotor areas in patients with CI in ALS [104] (Table 1). ALSci patients showed increased ReHo in bilateral inferior parietal lobules and the precuneus and inferior cerebellar areas, as well as in the right inferior parietal and right inferior cerebellar area relative to the ALS non-CI subgroup, while both ALS groups showed decreased ReHo in bilateral sensorimotor cortices. The altered regional functional coherence might indicate the underlying deficits in ALS with and without CI, supporting that ALS is a multi-system disease and providing evidence that alterations of ReHo in the right inferior cerebellar area might be a marker of ALSci [105].

In addition to a volume reduction in the basal ganglia [85,96,97], the atrophy of the amygdala [90] and the cholinergic nucleus accumbens was described in both ALS cases with and without CI [101,102,103]; the clinical implication of the latter is under discussion [102].

Recent MRI clustering revealed three ALS subtypes with specific neurodegeneration and clinical patterns. All subgroups displayed the involvement of the precentral gyrus and are featured, respectively, by (1) pure motor involvement, (2) orbitofrontal and temporal affection (frontotemporal cluster /FT/), and (3) involvement of posterior cingulate cortex, temporal operculum, parietal WM and cerebellum (cingulate-parieto-temporal cluster /CPT/). These subgroups showed distinct clinical profiles, among which FT and CPT revealed higher rates of CI— the FT had significantly higher frequencies of CI in language, EF and memory domains. Longitudinally, this clustering of subgroups remained stable: 90.4% of the patients remained stable at follow-up visits, clustered in the same subgroup as at baseline [100].

In conclusion, the cortical thickness of ALS patients is correlated with neuropsychological scores, which may reflect not only the cortical structure but also the regional homogeneity corresponding to the cognitive assessment and may provide help for the early diagnosis of cognitive changes in ALS patients [31]. The new clustering of ALS into three main patterns of neurodegeneration, each associated with distinct clinical manifestation and cognitive profiles, which remained stable during the disease process, designates a process in classifying this heterogeneous disorder based on characteristic morphological features.

## 4. Brain Perfusion and Metabolism Studies

PET studies showed reduced regional cerebral blood flow (rCBF) in the anterior cingulate cortex in both ALS with and without CI; ALS patients with impaired verbal fluency showed significantly reduced rCBF in the bilateral prefrontal cortex, rostral anterior cingulate, right parahippocampal gyrus and anterior thalamic nucleus. rCBF at rest in the right parahippocampal gyrus was significantly correlated with verbal fluency score, as was the activation of the anterior thalamic nucleus complex, indicating functional abnormalities in regions along a limbo-thalamo-cortical pathway [106]. 18F-FDG-PET showed moderately reduced frontal and prefrontal metabolism in ALSci, which increased with the severity of CI [107]. A combined MRI and PET study in patients with ALS found GM atrophy, predominantly in the temporal poles, and hypometabolism in the left superior medial cortex [95]. Hypermetabolism was also found in parts of the temporal lobes and the cerebellum. A series of negative correlations between cognitive performance and regional cerebral metabolism in functionally relevant areas suggest that hypermetabolism is more likely to reflect deleterious processes such as neuroinflammation rather than compensatory neuronal activity [86].

Structural neuroimaging revealed different patterns of GM and WM involvement in the hypo- and hypermetabolic brain regions in patients with ALSci; cortical thickness and surface area showed differential involvement in the hypo- and hypermetabolic regions [116].

In the ALS group, the metabolism in the medial frontal cluster positively correlated with that of frontotemporal regions, bilateral caudate nucleus, and right insula, and negatively with that of corticospinal tracts, cerebellum, and pons. The negative correlation between the medial frontal cluster and the cerebellum found only in ALS might reflect cerebellar compensation. In addition, there is a negative correlation between education and brain metabolism in the right anterior cingulate and bilateral medial frontal gyrus [117].

Brain metabolism was positively correlated with that of prefrontal, orbitofrontal and anterior cingulate cortices; it was lower in association with the APOE ε/2 allele, indicating its role as a risk factor for CI in ALS [110].

Magnetic resonance spectroscopy in ALS patients exhibited higher hippocampal total N-acetylaspartate (tNAA), tNAA/total creatine (tCr) and total choline (tChol) bilaterally, despite the absence of volumetric and perforant pathway zone diffusivity differences between ALS patients with and without CI. Furthermore, superior memory performance was associated with higher hippocampal tNAA/tCr bilaterally [118]. Other studies showed reduced N-acetylaspartate rates in the motor cortex at baseline, while NAA ratios were reduced in the prefrontal cortex in cognitively impaired ALS patients [119]. The anterior and superior tuberal subregions of the hypothalamus containing the paraventricular nucleus (housing oxytocin-producing neurons) display considerable volume loss in ALS and ALS-FTD, while the inferior tuberal subregions with the paraventricular nucleus (containing neuroendocrine neurons) were relatively preserved, supporting discrete neuropeptide expression abnormalities that may potentially underlie the pathogenesis of certain cognitive and behavioral symptoms in this disease spectrum [91].

## 5. Brain Network Studies

Brain network analyses have contributed valuable insights into cerebral pathology in vivo. Previous studies have reported conflicting data of increased, reduced or unaltered resting state networks (RSNs) and functional connectivity (FC) in ALS and did not report the contributions of upper neuron changes to RSN FC [120,121]. Impaired cognitive flexibility was related to GM atrophy in inferior frontal and insular regions, and to reduced FA in WM projections in the inferior fronto-occipital and uncinate fasciculi and corpus callosum [98]. Later studies suggested that executive dysfunction in ALS is associated with frontal and global network dysconnectivity, underlain by diminished WM integrity, which indicates domain-specific or general CI, depending on the degree of global network destruction [122]. Widespread disease-associated disruptions pointed to extensive dysfunctions in both motor and cognitive networks [123], with particular dysregulation and asymmetry of the hemodynamic-based frontal functional network [124].

In addition to multiple GM volume reductions in precentral gyri, posterior cingulate cortex, basal ganglia, thalamus and hippocampus, ALSci patients revealed the involvement of the superior longitudinal fasciculi and corpus callosum, reduced FC in the sensorimotor cortex—RSN and frontal pole in the bilateral thalamic-visual cortex network, and increased FC in the left primary motor cortex and left frontoparietal network [85]. There was a diffuse GM volume reduction in the primary motor cortex bilaterally, and the frontotemporal areas, cerebellum and basal ganglia were associated with decreased FA and increased radial diffusivity in the corticospinal tract bilaterally and in several WM tracts [83]. Diffusion tensor tractography detected a unique contribution of microstructural changes in hippocampal and frontotemporal WM tracts in ALS patients with impaired episodic memory [96]. ALSci patients demonstrated an “ALS with only motor deficits (ALS-cn)”-like pattern of structural damage, diverging from ALS-cn, with similar motor impairment for the presence of enhanced functional connectivity within sensorimotor areas, supporting the hypothesis that ALSci might be considered as a phenotypic variant of ALS, rather than a consequence of disease worsening [125]. Reduced GM in the left hippocampus, entorhinal cortex and posterior cingulate, as well as increased FA and decreased mean diffusivity in the left cingulum bundle (hippocampal part) was accompanied by decreased FC in the bilateral hippocampus, anterior and posterior parahippocampal and posterior cingulate gyrus, indicating widespread connectivity abnormalities in the Papez network [99]. While non-demented ALS patients without episodic memory deficits had no changes in the GM of the Papez circuit, those with delayed recall showed marked atrophy of the GM of the Papez circuit, indicating that it is differentially affected in ALS with and without cognitive changes [126]. Other studies revealed significant functional and structural connectivity changes across regions comprising the Papez circuit, and more extended areas including the cerebellum, frontal, temporal and parietal areas, supporting the theory of a multi-system pathology in ALS that spreads from cortical to subcortical structures [127].

Multimodal neuroimaging studies identified a significant volume reduction in the thalamus and putamen of non-fluent-variant primary progressive aphasia (nfvPPA) patients, while patients with semantic variant primary progressive aphasia (svPPA) only exhibited changes in the left hippocampus. The bulk of striatal and thalamic pathology in nfvPPA patients was identified in foci projecting to motor areas. Subcortical density alterations in svPPA patients were limited to motor areas. Striatal and thalamic changes indicated selected network-defined vulnerability patterns mirroring cortical pathology. Multimodal cortico-basal imaging confirmed that subcortical GM profiles of PPA phenotypes support the concept of network-wise degeneration, preferential vulnerability and disease propagation along specific connectivity patterns [128].

ALS individuals displayed a higher intra-network resting state functional connectivity in the sensorimotor, default mode, bilateral frontoparietal and orbitofrontal RSNs, underpinning both motor and cognitive impairment [129]. They showed an increased functional network, predominantly encompassing the connections between the default mode network (DMN) and the frontoparietal network (FPN). The increased structural connection involved interregional connections between the limbic network and the DMN, and the salient/ventral attention network and FPN, while the decreased structural connections mainly involved those between the limbic and the subcortical network. These FC changes are related to cognitive performance and predict the progression of disease [130]. The dysfunction of circuits connecting basal ganglia with the frontal lobe, in particular, sensorimotor and limbic circuits, contributes to cognition and motivation disorders [129,131] (Figure 1). A reduction in empathic skills, intention and emotional attribution are correlated with FA along the forceps minor, genu of corpus callosum, right uncinate and inferior fronto-occipital fasciculi; the involvement of frontal commissural fiber tracts and associative fronto-limbic pathways is the microstructural hallmark of the impairment of the high-order processing of socio-emotional stimuli in ALS [132].

CI in ALS, most commonly EF and language impairment, are associated with broad network dysfunctions in frontostriatal and frontotemporal systems [133], while altered decision-making showed a strong correlation with decreased connectivity in medial cingulate gyrus and the prefrontal area [134]. The letter fluency fMRI task revealed significantly impaired activation in the middle and inferior frontal gyri and anterior cingulate gyrus, in addition to regions of the temporal and parietal lobes; confrontation naming was associated with impaired activation in less extensive prefrontal regions, including the inferior frontal gyrus and regions in the temporal, parietal and occipital lobes. These findings further illustrate the heterogeneity of cognitive and cerebral changes in ALS [135]. Other studies showed that different executive deficits, like letter fluency, processing speed or dual-task performance, are related to distinct (inferolateral) prefrontal dysfunction [136]. Resting-state fMRI revealed increasing functional breakdown within the frontostriatal and frontoparietal networks related to EF frontal dysfunction [131,137]. Disease progression was associated with a frontal functional network disruption that was primarily observed in the right prefrontal cortex, suggesting that dysregulation, centralization and asymmetry of the hemodynamic-based frontal functional network are potential neurotopological markers of ALS pathogenesis [124]. Similar patterns of hypoconnectivity in the bilateral motor and frontotemporal cortices emerged when comparing ALSci and ALS-FTLD patients with those not cognitively impaired [138]. Diffusion tensor MR imaging that is sensitive to microstructural changes in WM tracts showed that attention and EF impairment correlated with changes in the corpus callosum and association WM tracts bilaterally, including cingulum, inferior longitudinal, inferior fronto-occipital, and uncinate fasciculi. Verbal learning scores were associated with fornix microstructural changes, and visuospatial abilities correlated with left uncinate FA, underlining the role of WM network abnormalities in the development of CI in patients with ALS [139].

In ALS, the inferior frontal gyri show significantly lower baseline activity compared to healthy controls; these activities correlate with motor decline and cognitive inhibition, indicating a symptom-preceding and -associated motor and cognitive cortical network decline [140]. Resting-state fMRI showed that FC within the DMN, as well as between the DMN, the sensorimotor network (SMN), the frontoparietal network and the salience network (SN), are predictive of both global cognition and motor function in ALS [141]. Moreover, the mental fatigue score is directly related to FC in the right and left insula (within the SN) and inversely related to that in the left middle temporal gyrus (within the DMN), thus correlating with CI due to alterations of FCs in extramotor networks [142]. Decreased FC was observed for the SMN and DMN in the left medial frontal and left postcentral gyrus, for DMN in the right medial frontal gyrus and left precuneus, for FPN in the right and left middle frontal and left inferior frontal gyrus, and for subcortical network in the right and left anterior insular cortex and anterior cingulate cortex, whereas the DMN showed increased FC in the middle temporal cortex [142] (Figure 2). Furthermore, some alterations of fronto-temporo-parietal-cerebellar circuits could be related to the pseudobulbar effect in ALS. In particular, abnormal FC between the cerebellum and posterior cingulate and the left middle frontal cortex indicates a crucial role of the cerebellum in regulating emotional expression in ALS [142].

In conclusion, brain network analyses revealed involvement of multiple FC dysfunctions related to CI in ALS, particularly affecting fronto-striatal, frontotemporal, and frontoparietal systems, the Papez circuit, fronto-temporo-parieto-cerebellar circuits, and DMN, SMN and SN, with the affection of different networks contributing to specific cognitive and motivation disorders. A schematic overview of the major disrupted architectures is shown in Figure 2 (see also Appendix A).

## 6. Neurophysiological Studies in ALSci

The evaluation of event-related potentials (ERP) in ALS patients with CI detected changes in the neural substrate of cognitive changes [143], confirming earlier studies of ERPs showing lower P3 (novelty P300) amplitudes, indicating a subclinical impairment of cognitive functions, which may be related to dysfunction of the frontal network [144]. The determination of cognitive profiles in non-demented patients with early-stage sALS by using ERPs revealed significantly prolonged latencies, indicating subclinical cognitive deficits in non-demented sALS patients [145]. A loss in performance in cognitively impaired ALS patients was accompanied by a decrease in the P300 event-related potential and a decrease in the task-relevant EEG band power [146]. In a similar way, reduced amplitudes and delayed latencies in attenuated error-related potentials may indicate ALS-associated impairment of executive functions, potentially due to disturbances in neural networks that involve the anterior cingulate cortex [147]. Auditory ERPs and delayed latencies were associated with attention impairment in ALS, supporting the hypothesis that ALS involves extramotor cognitive functions including auditory attentional processing [148].

Recent studies have shown that resting-state EEG can quantitatively capture abnormal patterns of cognitive network disruptions in ALS. These changes have been identified across multiple frequency bands by measuring neuronal activity (spectral power) and connectivity on source-located brain oscillations from high-density EEG. Based on data-driven methods (spectral clustering and similarity network fusion), a clustering analysis was undertaken to identify disease subphenotypes and to determine whether different patterns of network disruption may be predictive of disease outcome. According to these results, ALS patients can be subgrouped into phenotypes with distinct neurophysiological profiles. They are characterized by varying degrees of disruption in (1) the somatomotor (alpha-band synchrony) network, (2) the frontotemporal (beta-band neural activity and gamma-l-band synchrony), (3) frontoparietal (gamma-l-band co-modulation) network, which reliably correlate with distinct clinical profiles and different patterns of network disturbance. These data demonstrate that neuroelectric analysis can distinguish various disease subtypes based on different patterns of neurophysiological network disturbances, which may reflect the underlying disease neuropathobiology. Advanced network profiling in ALS may enable new therapeutic strategies that are based on neurobiology and designed to modulate network disruption [149]. In a similar way, the source localization of evoked potentials can reliably discriminate patterns of functional network impairment in ALS subgroups with dysfunctions of attention. The discriminating ability of the detected cognitive changes in specific brain regions (in particular, the left inferior frontal, dorsolateral prefrontal, superior temporal, and posterior parietal cortices) is comparable to those of fMRI and thus provides a novel non-invasive quantitative biomarker of network disruption in ALSci patients [150].

In conclusion, neurophysiological methods not only revealed early cognitive and executive dysfunctions in ALS but demonstrated different patterns of neural network disruptions, thus providing novel non-invasive quantitative biomarkers of network disorders related to cognitive disorders in ALS.

## 7. Neuropathological Findings

ALS is histologically characterized by large variety of cytoplasmic inclusions in both neuronal and glial cells positive for transactivation response DNA-binding protein 43 kDa (TDP-43); this is a key feature for up to 97% of ALS cases [15,16], whereas others show FUS and/or tau reactivity (see [151,152]). TDP-43 and FUS as the defining pathological hallmarks in ALS, coupled with ALS-causing mutations of both genes, indicate that their dysfunction damages the motor system and cognition [153]. In sALS, a model of the staging of TDP-43 has been proposed: the involvement of the agranular motor cortex and alpha-motor neurons of the brainstem and spinal cord (stage 1), in the prefrontal neocortex (middle frontal gyrus), reticular formation and the precerebellar nuclei (stage 2), in further areas of the prefrontal neocortex (gyrus rectus, orbitofrontal gyri), postcentral sensory cortex and basal ganglia (stage 3), and in anteromedial temporal lobe including the hippocampus (stage 4). Based on this staging, a corticofugal model for a “prion-like” spreading of TDP-43 pathology was hypothesized suggesting that it starts from the primary motor cortex and spreads from there via axonal projections in a spatiotemporal manner to lower motor neurons and to subcortical structures [154,155,156,157]. TDP-43 is self-templating and -propagating across the brain, due to the presence of a glycine-rich C-terminal domain that is prone to aggregation and misfolding; this induces the formation of TDP-43 oligomers and invariably reduces cell viability [158,159]. Moreover, TDP-43 can transport intracellularly in an anterograde and retrograde fashion, supporting the region-to-region spread of TDP-43 pathology [160].

From the clinical standpoint, ALS patients affected by extramotor deficits including cognitive disturbances seem to correspond to the pathological involvement of the relevant cerebral structures [161]. Loss of TDP-43 from the nucleus and the subsequent aggregation in the cytoplasm, occurring in susceptible regions may be associated with neuronal loss, although this may also occur without TDP-43 accumulation. Therefore, it remains controversial whether the stepwise appearance of TDP-43-positive inclusions is based on direct cell-to-cell propagation [152]. In C9orf72 repeat mutations, the most common genetic impairment causal to ALS, early cortical network dysfunction is due to impaired synaptic plasticity, attributable to impaired pre-synaptic vesicle dynamics, which directly impact cortical function [162].

Under the assumption of a single common progression pattern, a recent staging of TDP-43 proteinopathies using SuStaIn classification was proposed. The SuStaIn stage can be thought of as a proxy for progression along a pathological trajectory [163,164]. SuStaIn estimated that TDP-43 deposition in ALS began in the spinal cord, before progressing to the medulla and motor cortex (stages 2–5). Subsequently, in stages 6–13, there was TDP-43 deposition in the caudate/putamen, thalamus, globus pallidus, midbrain, substantia nigra, pons and anterior cingulate cortex. In stages 14–15, it progressed to the middle frontal and angular gyri, and by stage 21, TDP-43 was found in all regions except the cerebellum, including the medial temporal lobe [165]. In addition, ALS subtypes were assessed for differences in demographic, pathological and genetic variables. Two confounding factors significantly discriminated between two subtypes: ALS subtype 1 (subcortical predominant) has the tendency to have less overall brain pathology (e.g., lower SuStaIn stage) compared with ALS stage 2 (corticolimbic predominant), with the onset of motor cortex pathology. SuStaIn inferred the spinal cord as the first susceptible region for both ALS subtypes [165]. Studies to identify molecular targets in cognitively affected and unaffected brain regions in ALS patients, using sensitive mRNA sequencing, identified 50 significantly dysregulated genes that are distinct between these brain regions. Using in situ hybridization and macromolecular complex regulation, notably, NLRP3 inflammasome modulation was found as a potential pathological correlate of cognitive resilience in ALS [166].

Several studies showed that TDP-43 pathologic burden in ALS is associated with CI, though no association with disease duration or rate of progression was seen [62,167]. However, TDP-43-positive neuronal and glial inclusions were more numerous in ALSci than in classical ALS [168], and a close relationship between cognition and the extent of TDP-43 pathology was seen in non-primary motor areas [169]. ALS patients with impaired EFs showed TPD-43 pathology as well as significant neuronal loss and significant microglial activation in the middle frontal and superior or middle temporal gyrus [101]. TDP-43 burden in the limbic system (amygdala, dentate nucleus, CA1 sector of the hippocampus, subiculum, and entorhinal cortex) was greater in older than in younger ALS patients, indicating that the amygdala and hippocampus are vulnerable to TDP-43 pathology in older ALS patients [170]. Bulbar ALS cases showed severe neuronal loss and TDP-43 pathology across most speech regions, inducing a link between the severity of bulbar ALS and speech network damage, whereas spinal onset ALS cases showed no neuronal loss but mild TDP-43 pathology in focal regions, and cases without ante mortem bulbar ALS demonstrated an absence of pathology [171].

Other pathology studies showed that all ALSci patients had TDP-43 pathology in extramotor brain regions, associated with disorders of executive, language and fluency domains, although there was a subgroup with no cognitive dysfunction, despite having substantial TDP-43 pathology [27]. Postmortem studies in a large cohort of ALS cases, 38.5% of which had cognitive decline, revealed TDP-43 pathology and hippocampal sclerosis. Mutation carriers presented a higher burden of TDP-43 pathology than sporadic cases. Moreover, 89% presented some degree of concomitant pathologies, which were associated with older age at death, indicating that other findings can influence cognitive status, particularly in older age groups [172], although no cognitive burden from vascular risk factors was found in patients with ALS [173].

## 8. Cognitive Reserve in ALS

Long-time life experiences, such as education, occupational attainment, leisure activities, and bilingualism, have been considered proxies of cognitive reserve (CR). In neurodegenerative diseases, CR is considered a modulator of more favorable cognitive trajectory and motor functions. Cross-sectional studies and longitudinal assessment in cohorts of ALS patients measured CR by combining education, occupational attainment, amount of leisure activation, physical activity, etc. Jobs requiring greater reasoning abilities, social skills and humanities knowledge were related to preserved cognitive functioning consistent with CR [174]. It was hypothesized that higher CR would correlate with better performance on cognitive test batteries. The analyses provided moderate to strong evidence that higher CR was associated with a mild increase in performance with regard to verbal fluency functions, working memory, verbal learning and recognition, and visuoconstructive ability. These results indicated that CR moderates the effects of brain morphology on cognition in ALS. Executive functions presented a dissociation: while verbally assessed functions benefit from CR, non-verbally assessed functions did not [175]. Thus, CR was a significant predictor of baseline neuropsychological performance, with high-CR individuals performing better than those with medium or low CR. Better cognitive performance in high-CR individuals was maintained longitudinally for social cognition, EF or confrontational naming. High-CR patients displayed little cognitive decline over the course of follow-up, which suggests that CR plays a role in the presentation of CI at diagnosis but is not protective against cognitive decline [176]. However, according to others, there is evidence that CR protected letter fluency from further decline [177]. Another study documented the association between CR and cognitive performance in all patients and the predictive role of CR in modulating the extent of cognitive decline and functional bulbar impairment, suggesting that the concept of reserve applied to ALS should encompass cognitive but also motor domains [56].

Usually, there is an association between the pathological burden of TDP-43 misfolding and CI in ALS, demonstrating high specificity, but correspondingly low sensitivity characterizes a subset of individuals with no evidence of cognitive deficits despite the high burden of TDP-43 pathology—called “mismatch” cases. These cases have demonstrated predominantly the expression of clusterin, a topologically dynamic chaperone protein with the ability to participate in both intra- and extracellular proteostasis. Mismatch cases showed a differential spatial expression of clusterin, with a predominantly neuronal pattern, compared to cases with a cognitive manifestation of their TDP-43 pathology, which demonstrated a predominantly glial distribution. These data indicate that in individuals with TDP-43 pathology, a predominantly neuronal expression of clusterin in extramotor brain regions may indicate a cell protective mechanism, delaying the clinical consequences of TDP-43 pathology such as cognitive dysfunction [178].

Another key molecular target underlying cognitive resilience in ALS is NLRP3 inflammasome modulation, which might be a potential therapeutically targetable correlate of cognitive resilience in ALS [166]. However, future research is needed to examine the interaction between CR and other objective correlates of CI in ALS.

## 9. Experimental Models of ALSci

Experimental animal models are intended to provide information about the mechanisms and the brain regions that are involved in the alterations occurring in CI in ALS.

Dominant missense mutations in TDP-43 cause up to 95% of ALS, and the cytoplasmic accumulation of TDP-43 is a pathological hallmark in ALS. Investigations of the transgenic (tg) mouse model expressing the disease-causing human TDP-43 M337V mutant have shown robust motor and cognitive deficits in hemizygous mice by 8 months of age. After 12 months, cortical neurons were significantly affected by cytoplasmic TDP-43 mislocation, mitochondrial dysfunction, and neuronal loss. Compared with age-matched non-tg mice, TDP-43 M337V mice showed a similar expression of TDP-43 but higher levels in mitochondria, while an inhibitory peptide abolished cytoplasmic TDP-43 accumulation, restored mitochondrial function, prevented neuronal loss and alleviated both motor and cognitive deficits [179]. A tg mouse model overexpressing human wild-type TDP-43 protein showed a rapid development in cognitive/social deficits involving working and recognition memory in the absence of overt motor abnormalities [180]. In a mouse line expressing an ALS-linked mutation in TDP-43 (Q331K), motor and cognitive deficits were observed at 3 months of age, persisting through 12 months of age. Among the cognitive modalities, the hippocampus-mediated spatial learning and memory were normal, whereas the frontal-mediated working memory and cognitive flexibility were impaired. Q331K TDP-43 was largely retained at the nucleus without apparent aggregates, which was comparable with the physiological level of human TDP-43 protein in cerebral cortex and hippocampus. These data suggest that the motor and frontal cortex are more vulnerable to disease-linked mutation in TDP-43 [181]. A recently developed C9orf72 BAC tg mouse model showed the widespread loss of H3K9me3 in astrocytes and neurons in the spinal cord, motor cortex, and hippocampus, associated with hippocampal-dependent CI [182].

SOD1 mice, a model for fALS, showed a reduction in dendritic length and branch nodes of basal dendrites in the pre/infralimbic medial prefrontal cortex neurons, indicating abnormal prefrontal connectivity and function before the onset of motor disturbances [183], while another SOD1 mouse model showed CI due to disordered cortico-striatal and hippocampal synaptic plasticity, dendritic branching and glutamate receptor function [184]. A mouse model expressing wild-type human FUS (found in sALS carrying FUS mutations) developed hippocampus-mediated cognitive deficits, accompanied by an age-dependent reduction in spine density and long-term potentiation in their hippocampi. However, there were no definite FUS aggregates or cytosolic FUS accumulation. It was found that FUS directly binds to Sema5a (a gene involved in axon guidance and spine dynamics) mRNA and regulates Sema5a expression in a FUS-dependent manner, suggesting that FUS-driven Sema5a deregulation may be related to cognitive deficits in FUS tg mice [185]. Another mouse model expressing an ALS-linked human FUS mutation (R514G-FUS) developed cognitive deficits accompanied by a reduction in spine density and long-term potentiation within the hippocampus, accompanied by a disruption in protein homeostasis and mitochondrial functions, a widespread reduction in cortical volumes, and enhanced FC between the hippocampus, basal ganglia and neocortex. These findings suggest that disease-linked mutation in FUS may lead to changes in proteostasis and mitochondrial dysfunction that affect brain structure and connectivity, resulting in cognitive deficits [186].

In conclusion, various tg animal models of ALS have provided insight into the heterogeneity of brain lesions associated with cognitive dysfunction, into and the role of gene mutations in the development of cognitive disorders in ALS.

## 10. Putative Pathogenic Mechanisms

ALS is a multi-system neurodegenerative disorder with both clinical and biological heterogeneity for which the dysregulation of several pathogenic mechanisms related to TDP-43 pathology has been postulated, including disorders of protein homeostasis, RNA processing, nuclear and axonal transport, energy metabolism, dysfunctional immune regulation, oxidative stress, excitotoxicity, endosomal and vesicular transport impairment, autophagy, neuroinflammation involving glial cells and circulating immune cells, and mitochondrial dysfunction [16,23,151,187,188,189,190,191]. On the other hand, evidence for the primary dysfunction of mitochondrial oxidative phosphorylation was detected and provided in vivo evidence for bioenergetic dysfunction in ALS brains as a central pathogenic factor [190]. Disease-linked mutations in FUS may lead to changes in proteostasis and mitochondrial dysfunction that in turn affect brain structures and connectivity, leading to cognitive deficits [186]. The molecular background of protein aggregation in the pathogenesis of ALS was reviewed recently [192]. In addition to oxidative stress [193], there is recent evidence that the endoplasmic reticulum stress pathway is an essential pathological mechanism in ALS [194].

Electron microscopy analyses of ALS patient brain samples revealed prominent mitochondrial impairment, confirming that increased TDP-43 expression induced mitochondrial dysfunction and elevated the production of reactive oxygen species. TDP-43 expression depressed mitochondrial complex I activity, reduced mitochondrial ATP synthesis, and activated the mitochondrial unfolded protein response, though it down-regulated mitochondrial protein, increased mitochondrial TDP-43 levels, and exacerbated mitochondrial damage as well as neurodegeneration. These data demonstrated that TDP-43-induced mitochondrial impairment is a critical pathogenic factor in ALS and other TPD-43 proteinopathies [195].

There is growing evidence for a relationship between TDP-43 and glial pathology, indicating pathogenic glial cell-autonomous dysfunction and dysregulated glial immune responses to neuronal TDP-43 [196]. Serum levels of glial fibrillary acidic protein provide neurochemical evidence of astrocyte involvement in ALS pathophysiology, particularly in the development of CI [197]. However, the relationship between TDP-43 and neurodegeneration is not absolute, and other pathophysiological processes, such as neuroinflammation (with the prominent role of microglia, cortical hyperexcitability, and synaptic dysfunction), play a central role in ALS pathophysiology [189]. There are links between the C9orf72 repeat expansion and neuroinflammatory signatures that exist across genetic and sporadic ALS cohorts [198]. Cortical hyperexcitability, probably associated with TDP-43 accumulation, was more prominent in cognitively impaired ALS [199].

Increasing evidence suggests that synaptic dysfunction is a central and possibly triggering factor in ALSci [17,18,200,201,202]. Cognitively impaired ALS cases showed a relation between TDP-43 pathology and synapse loss in the frontal cortex, which was not due to cortical atrophy nor associated with other dementia-related neuropathologies, indicating that synapse loss in the prefrontal cortex represents a neurobiological substrate for cognitive decline in ALS [200]. The impaired clearance of TDP-43 and synaptic alterations may induce neuronal dysfunction related to TDP-43 pathology [203]. In addition to TDP-43 mislocation or aggregation as a hallmark of ALS, there is evidence for alterations in the metabolism of the microtubule-associated protein tau, which is characterized by pathological phosphorylation at residue threonine 175, and the formation of cytoplasmic inclusions. Immunohistochemical studies using polyclonal phospho-tau antibodies and antibodies directed against PHF tau detected tau-positive intraneuronal and neuritic aggregates throughout the amygdala, the entorhinal, anterior cingulate and superior frontal cortices, and the substantia nigra, as well as immunoreactive tufted astrocytes in the same regions in ALSci brains. The extent of pathological tau inclusions showed no correlation with that of TDP-43 pathology, and nuclear TDP-43 immunoreactivity was absent in neurons with tau pathology. These findings indicate that the ALSci brain is characterized by the co-occurrence of TDP-43 and tau pathology [204]. Other studies corroborated this hypothesis—for example, increased TDP-43 accumulation and lethality were observed in an in vivo model co-expressing both proteins [205]. The same group also demonstrated that low levels of TDP-43 already promote neurotoxicity, leading to selective neurodegeneration [206]. These data indicate that TDP-43 may promote tau aggregation [207,208]. The interaction between both proteins suggests that they share pathomechanistic events that exert a synergistic role in ALSci. A potential role of TDP-43 acting synergistically with pathological tau metabolism has been proposed [209] and demonstrated in a rodent model [210].

On the molecular level, TDP-43 and FUS participate in the biogenesis and metabolism of coding RNAs and in the transport and translation of mRNA as part of cytoplasmic mRNA-ribonuclein, many of which are involved in synaptic transmission and plasticity, indicating that synaptic dysfunction could be an early process contributing to motor and cognitive deficits in ALS [153]. In vivo studies of synaptic density as a possible biomarker with 18F-SynVesT-1 PET also showed reduced synaptic density in the bilateral superior temporal gyrus, hippocampus-insula, anterior cingulate, and left inferior frontal gyrus in ALSci patients compared to healthy controls, which could be a potentially useful biomarker for ALS and for estimating cognitive decline [211].

Synaptic proteomics experiments found more than 30 ALS-associated proteins in synaptoneurosomes, including TDP-43, FUS, SOD1 and C9orf72. Stratifying the ALS cohort by cognitive status revealed almost 500 specific alterations in cognitively impaired ALS synaptic preparations, thus generating an unbiased map of the human ALS synaptic proteome, which highlighted the influence of cognitive decline and C9orf72-repeat expansion on synaptic composition [212].

In conclusion, a complex multi-step process of events associated with TDP-43 and other mechanisms is engaged in the development of motor and cognitive disorders in ALS. Among these, besides other alterations, mitochondrial and synaptic impairment and proteasomic dysfunction represent the main mechanisms underlying ALS etiopathogenesis. Functional and structural events probably represent important mechanisms underlying an adaptive capability, causing the partial and transient resiliency of the CNS affected by neurodegeneration, while on the other hand, the failure of synaptic functions and plasticity are part of an early pathogenic process in ALS responsible for cognitive dysfunction [213].

## 11. Conclusions and Outlook

ALS is a multi-system neurodegenerative disorder with clinical and biological heterogeneity that is accompanied by a wide spectrum of cognitive decline, covering various domains. The pathogenesis of ALS and the accompanying cognitive dysfunctions include heterogeneous mechanisms, dominated by the mislocation and aggregation of TDP-43, which is associated with a wide variety of mechanisms, including disorders of proteostasis, mitochondrial dysfunction, oxidative stress, neuroinflammation and immune dysregulation. This leads to the neurodegeneration of essential cerebral systems, with synaptic dysfunction as a central and triggering factor of CI in ALS and related FTLD. Modern neuroimaging techniques revealed the widespread degeneration of fronto-limbic systems with the microdestruction of GM and WM systems, hypometabolism in frontotemporal regions, and the disruption of frontoparietal, frontostriatal and other networks as essential markers of cognitive deficits. Neuropathology confirmed the central role of TDP-43 burden, related neurodegeneration and synapse loss in frontal and other cognition-relevant cerebral regions. ALS is characterized by the dysfunction of specific neuronal networks and alterations of synaptic connectivity, contributing to neuronal degeneration, which leads to the impairment of motor and cognitive functions [214]. The role of cognitive reserve has become the focus of modern neuroscience and of the discussion of the diagnostic and prognostic value of modern biomarkers in ALS and their potential to reveal insights into the pathophysiology of CI in ALS. Emerging imaging biomarkers of extramotor neurodegeneration that enable the monitoring of disease progression are particularly promising. In addition, a growing arsenal of biofluid biomarkers linked to neurodegeneration and neuroinflammation as well as to synapse dysfunction and neuronal demise are important for improving the diagnostic accuracy and identification of patients with a faster progression of both motor and cognitive dysfunctions.

Neurofilaments, indicating ongoing neuronal or axonal injury, are one of the most promising biomarkers of ALS, although their role in the pathophysiology of ALS is unclear [215]. A significant elevation of the neurofilament light chain (NfL) in cerebrospinal fluid (CSF) and serum was found in ALS patients, compared to healthy controls. Increased values were found particularly in ALS patients with known mutations but have also been applied to sALS patients and may shorten the diagnostic delay by up to 3 months [20]. ALS is characterized by increased CSF and serum levels of NfL, a marker of neuroaxonal degeneration [216,217] that may be associated with CI [218]. On the other hand, the finding of moderate serum NfL elevation in patients with a long ALS duration underlined its value as a progression marker [219]. Sensitivity and specificity values of NfL in CSF have been reviewed recently [220]. Since inflammatory processes involving astroglia and peripheral immune cells are also characteristic in the pathophysiology of ALS, proinflammatory biomarkers may provide information on the disease stage, progression rate and pathophysiological mechanisms [220].

The development and validation of biomarkers that detect the pathogenic aggregates of TDP-43 and related pathological proteins in vivo are notably expected to further elucidate the pathophysiological mechanisms underlying the basic disorder and the ensuing cognitive decline. Novel biomarkers tracking the different aspects of ALS pathophysiology and the mechanisms of related CI are notably expected to further elucidate the different aspects of ALS pathogenesis and to pave the way to precision medicine approaches in the ALSci continuum.

## Figures and Tables

**Figure 1 ijms-24-14647-f001:**
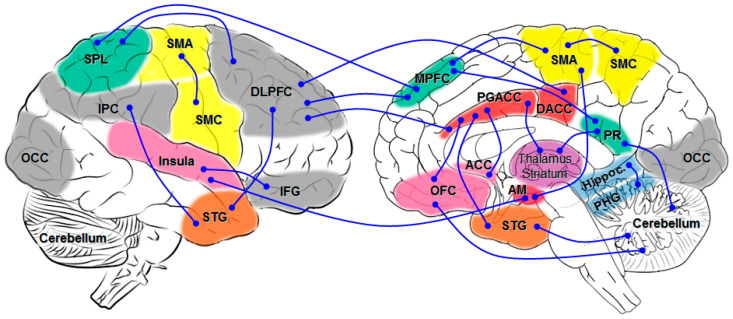
Schematic overview of major network connectivity in ALSci changes. AM amygdala; DACC dorsal anterior cingulate gyrus; DLPFC dorsolateral prefrontal cortex; IFG inferior frontal gyrus; IPC inferior parietal cortex; MPFC medial prefrontal cortex; OCC occipital; OFC orbital prefrontal cortex; PGACC perigenual anterior cingulate cortex; PHG parahippocampal gyrus; PR precuneus; SMA supplementary motor cortex; SMC sensorimotor cortex; SPL superior parietal lobule; STG superior temporal gyrus.

**Figure 2 ijms-24-14647-f002:**
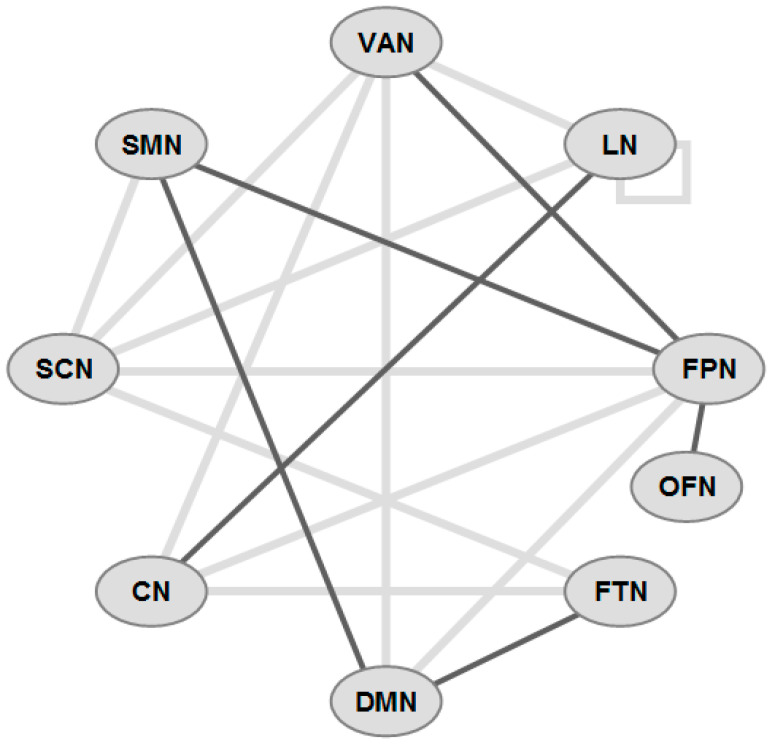
Schematic overview of functional connectivity (FC) in some major networks in ALS-CI (incomplete). Gray lines: decreased FC, black lines: increased FC. Network abbreviations: VAN ventral attention; LN limbic; FPN frontoparietal network; OFN orbitofrontal; FTN frontotemporal; DMN default mode; CN cerebellar; SCN subcortical; SMN sensorimotor.

**Table 1 ijms-24-14647-t001:** Major neuroimaging findings in ALSci.

Type of Lesion, Location	References
Cortical atrophy	
Frontoparietal cortex, somatosensory area	[87]
Frontal, temporal, and limbic cortex	[80]
Frontal, temporal and parietal cortex	[82,92]
Precentral gyri, posterior cingulate cortex, thalamus, striatum, pallidum, hippocampus	[85]
Left inferior temporal and anterior temporal cortex	[93]
Primary motor cortex and perisylvian regions	[94] (speech deficits)
Temporal poles	[95]
Inferior frontal, temporal, cingulate, and insular cortices, frontoparietal cortex	[84]
Frontoparietal cortex, right insula, striatum, precentral cortex, right frontal and temporal cortex, precuneus	[89]
Primary motor cortex, frontotemporal cortex, basal ganglia, cerebellum	[96,97]
Inferior frontal cortex, insular region	[98]
Left hippocampus, entorhinal cortex, posterior cingulate cortex	[99]
Precentral gyrus, orbitofrontal and temporal cortex, posterior cingulate cortex, temporal operculum, parietal white matter, cerebellum	[100]
Left entorhinal cortex -”- medial orbitofrontal lobe -”- inferior temporal gyrus -”- insular lobe	[31]
Amygdala	[90]
Nucleus accumbens	[101,102]
Right nucleus accumbens, left caudate nucleus and hippocampus	[103]
Reduced fractional anisotropy/altered regional homogeneity (ReHo)	
Corticospinal tract bilaterally Corpus callosum, extramotor tracts	[83]
Increased ReHo Left prefrontal lobe, anterior cingulate cortex	[104]
Increased ReHo: bilateral inferior parietal lobules, precuneus, right parietal and inferior cerebellar area Decreased ReHo: bilateral sensorimotor cortices	[105]
Reduced brain perfusion, hypometabolism	
Prefrontal cortex, anterior cingulate cortex, right hippocampus, anterior thalamic nucleus	[106]
Frontal and prefrontal area	[107]
Dorsomedial/dorsolateral prefrontal cortex	[108]
Left parahippocampal gyrus	[109]
Prefrontal, orbitofrontal, anterior cigulate cortex	[110]

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
