# Peer review of "The Spectrum of Cognitive Dysfunction in Amyotrophic Lateral Sclerosis: An Update"

_ijms, 2023, doi:10.3390/ijms241914647_

Round 1
Reviewer 1 Report
The author well review the issue of cognitive dysfunction in amyotrophic lateral sclerosis.
I hope the author to figure out the "Putative pathogenic mechanisms" if it is possible.
Do ALS patients have cognitive dysfunction? What kinds of cognitive dysfunction do the patients have?
This manuscript timely addresses the clinical question about the cognitive function in ALS.
I think this manuscript comprehensively covers the relevant papers.
I think there are no methodological problems.
It would be better to understand their putative mechanism of cognitive dysfunction in ALS.
Author Response
Many thanks for your kind comments!
The putative pathogenic mechanisms of cognitive dysfunction in ALS are now slightly revised.
Reviewer 2 Report
Abstract:
Gives a good overview of cognitive dysfunction in ALS but does not explicitly state what the reader can expect in the review. Such a statement would be good to add.
Introduction:
Lane 31: “ALS is of unknown etiology” This statement is difficult as several genetic factors are known and indeed mentioned by the author later in the introduction.
Lane 34: “Survival time is 3-4 year” Sentence should be specified. Three to four years based on first symptoms or diagnosis?
Lane 47: “Neuropsychiatrc disorders….” This sentence needs clarification as it is not clear who shows a “poorer overall cognition, visuospatial or language disorder,…” the parents or the ALS patient. Furthermore the start is strange as the “and their families” refers to the parents and not the ALS patients. Please rephrase.
Lane 53: Two parts of the sentence don’t fit together.
Lane 69: Also here, some more details of what can be expected in the next chapters would be good to mention.
2. Cognitive abnormalities in ALS
Lane 72: Cognitive involvement in what?
Lane 78: Here, a CI incidence of 30%-75% in ALS is mentioned while in the abstract a frequency of 27% to 75% is stated. Numbers should be the same, correct? Phrasing of the whole chapter about CI incidence needs to be specified: it is mostly not mentioned what the percentage is referring to, such as 50% of what? ALS patients?
Lane 91: “did not” needs to be changed to “does not”.
Lane 96: “were” needs to be changed to “are”…please check whole manuscript for this change in tense. (Lane 120, 122, 125, 128, 138 …and so on)
Lane 128: Why is hyposmia mentioned here? How does it relate to CI?
Lane 141-146: “ALS motor and cognitive components may worsen in parallel” this statement seems to contradict information provided Lane 109 +/-.
Lane 152: what kind of patients were compared? I don’t understand the sentence.
Lane 157: who is “some” and “others”? same in lane 163
Lane 157 and proceeding: It is here the third time the author refers to the relationship between CI and disease stage. All three should be merged.
Lane 166: a gene repeat is not a phenotype, it causes the phenotype.
Lane 169: who or what is “their”?
Lane 170: please explain what “ALS type 8 patients” are
3. Neuroimaging findings in ALS-CI (Table 1)
Table 1 should be mentioned in text, not in title.
In this chapter “over time” is used more than once. This expression is not very specific and should be changed to “during disease progression” or similar.
Lane 193: why are some brain regions mentioned in singular and others in plural? I have never heard of “thalami” either “thalamus” or ”thalamic nuclei”.
Lane 196: Sentence needs to be turned around. Cognitive changes are highly dependent on cortical atrophy, not vise versa.
Lane 200: “involvement of basal ganglia” in what kind? Pathology? Atrophy?
Lane 202: what is “caudate”?
Table 1: Menke 2017: pallidum, Bueno, 2018: posterior cingulate cortex? Bede 213: left caudate what? Kew, 1993: anterior cingulate cortex?
Table 1: sometimes it is mentioned that changes are “bilateral”. This implies that other changes are unilateral. Is this correct? I would suggest removing all “bilateral” and only mention what is reported to be “unilateral”
Text about Table 1 also needs to be checked for phrasing and specificity of sentences.
Lane 244: Shortly explain “dysexecutive syndrome”
Lane 255: What are these subgroups? Either explain subgroups or don’t mention that their exist.
Lane 272: Patients remained stable for how long? When was follow-up performed?
4. Brain perfusion and metabolism studies
Lane 309-311: Sentences make no sense. What is e/2 allele? ApoE? Please provide more information.
Lane 316: was this also in the hippocampus?
5. Brain network studies (Table 2)
Table 2: Menke et al.: thalamic or thalamus? Also in text lane 338
Table 2 and lane 348 and 384: posterior cingulate what? Cortex?
Lane 405: What kind of ALS, with CI or without? What are the controls? Healthy individuals?
6. Neuropathological findings
Lane 429: “…damage the motor system and cognition” this reads really odd.
Lane 439-441: there is too much “involved” in this sentence that makes it hard to understand.
Previous comments also apply to this chapter (anterior cingulate? And so on)
Lane 480: what is meant by “showed a trend between…”?
7. Cognitive reserve in ALS
Lane 486: occipital attainment or occupational attainment?
Lane 534: NLRP3 is here mentioned for the first time. Shouldn’t it be also mentioned in chapter 6. Neuropathological findings?
8. Experimental models of ALS-CI
A short introduction to this chapter would help, that states why animals models are discussed here. It should also be explicitly mentioned that only models with CI are discussed and that these are examples and probably not a complete list. The conclusion should also state what these models might be good for.
9. Putative pathogenic mechanisms
Lane 634: 18F-SynVesT-1 is the name of the radioligand. What would be the biomarker?
What is the difference between this chapter and chapter 6? Both describe the neuropathology of ALS, so why two separate chapters?
10. Conclusions and outlook
Nice!
In this chapter, biofluid biomarkers are mentioned in detail, although they are not discussed in any previous chapter. It would be nice to see such a chapter, e.g. discussing NFL and other biofluid markers.
General comments:
It would improve the manuscript if patients ALS group names (with and without CI) are unified throughout the manuscript. Currently, it seems that the author used the nomenclature of each cited publication. Often “in ALS” is used. Does this relate to all ALS patients, with or without CI? This is often not clear.
Text about neuroanatomical changes (chapter 3-5) needs to be checked for grammar, singular/plural, and, as already mentioned bilateral vs. unilateral, should be simplified.
Author should think of restructuring chapters. It might help to start on molecular levels (chapter 6 and 9) and proceed with chapter 2 and so on.
Instead of Table 1 and 2 that are quite redundant as content is described in main text, an illustration of neuroanatomical, metabolic and network findings would be nice to have as it would support the understanding of the detailed content.
Editing by native speaker required. Some mix of BE and AE.
Author Response
Many thanks for your meticulous and detailed comments! All of them have been considered and are answered as follows:
Most of your comments referring to specific lines are now implementd.
L 31: Statement is now omitted.
L 34: Survival time from diagnosis.
L 47: Sentence is now reorganized.
L 53: Sentence is now corrected.
L 69: Details are now added.
L 72: Cognitive "impairment" - corrected.
L 78: Differences corrected.
L 91: Corrected.
L 96: Corrected.
L 128: Hyposmia is now deleted.
L 141-146: Redundant sentences omitted.
L 152: Corrected.
L 157: Repetition omitted.
L 157: Omitted.
L 166: Corrected.
L 169: Corrected.
L 170: ALS type 8 is now explained.
"over time" was replaced by "during disease progression".
L 193: Thalami was replaced by thalamic nuclei.
L 196: Corrected.
L 200: Corrected.
Table 1: Corrections implemented. "bilateral" is correct according to Hu et al 2020.
L 244: "dysexecutive syndrome" replaced by "executive dysfunction" was corrected.
L 255: Clustering of subgroups was corrected.
L 272: Corrected.
L 309-311: Sentence corrected.
L 316: Yes, also in hippocampus - corrected.
L 405: inserted "healthy" controls.
L 429: Sentence corrected. .
L 439-441: Repeated "involved" replaced by affected.
L 480: Corrected.
L 486: Corrected.
L 534: NLRP3 is now also mentioned in section Neuropathology.
Chapter 8. Experimental models: Short introduction added.
L 634: Corrected.
Overlap between chapters 6 and 9: Chapter 9 (now Chapter 10) is now shortened and only discusses pathogenesis.
With regard to your suggestions concerning the putative pathogenic mechanisms, this chapter has been reorganized in order to avoid duplication with the neuropathology chapter.
In "Conclusions and outlook", reference to NfL and inflammatory biomarkers has been added.
Table 2 is now corrected and, partly following your concern and that of reviewer #3, is now removed from the main text and will be submitted as supplement. Instead, an illustration of the major neuronal networks affects is now added.
The proposed complete restructuring of chapters has not been peformed, since I want to present the clinical findings before discussing morphology and pathogenesis.
Reviewer 3 Report
This is an excellent, comprehensive and well-written review of a difficult subject- cognitive impairment (CI) in amyotrophic lateral sclerosis (ALS). The author is a recognized authority in human neurodegenerative diseases and disease syndromes (i.e., vascular dementias) and has presented ALS as the clinically, pathologically, imaging and molecular genetic heterogenous systems degeneration that it is. This review will stimulate investigations into many different avenues of ALS research. Most importantly, this review will move ALS solidly into the family of systems degenerations where it rightly belongs. It is no longer exclusively a disease of motor neurons, although that characteristic is how it is diagnosed, followed and serves as the major source of mortality in this otherwise rare condition.
The author presents much recent evidence from brain imaging and network studies that demonstrate the heterogeneity of ALS presentations and progressions. There is a comprehensive presentation of known autosomal gene mutations causal to familal ALS (fALS) that also play varying roles in their non-mutated forms in sporadic ALS (sALS). Here the author is cautious in assigning causality to gene presence, even in aggregated form (i.e. TDP43).
Overall this is an outstanding review of contemporary thinking about ALS and varying CI subtypes. Throughout the review the author "drops hints" about how CI subtypes may correlate with specific radiological patterns of degeneration, and these early findings will likely expand in the near future as ALS takes on more of the systems degeneration mantle that it truly is.
I'm not sure if the two enclosed Tables should continue to exist in the main text body, or whether they could be presented as Supplemental Tables. I can see arguments for both situations and defer to the author's preference.
Author Response
Thank you for your positive comments.
Partly following your proposal concerning the tables, Table 2 is now removed from the paper itself and will be submitted as a supplement, and a figure illustrating the major involved networks is now included.
Reviewer 4 Report
In this review article, Jellinger reports the state-of-the-art about current findings on cognitive impairment in amyotrophic lateral sclerosis (ALS) as provided by neuroimaging and neuropathological studies. More in detail, the author focuses on the role of wide-spread functional disruptions of motor and non-motor brain networks in the pathophysiology of cognitive deficits in ALS and its variants.
I found the paper interesting since cognitive impairment and non-motor symptoms may be crucial for the early detection of disease and also for the assessment of disease progression. I have only minor concerns about the topic and the paper's organization.
- It would be interesting to report on specific valuable neurophysiological studies, including those based on EEG and evoked potentials, concerning cognitive impairments and behavioural and psychiatric symptoms (BPSD) in ALS.
- It would be relevant to discuss whether specific genetic variants of ALS may express specific cognitive impairments.
See above.
Author Response
Thanks for your positive evaluation and for your proposals!
A special part on EEG and evoked potentials is now added. Cognitive impairments in genetic variants is now briefly discussed.
Round 2
Reviewer 2 Report
The manuscript is now significantly improved.
In 11. Conclusions and outlook, the author states: "ALS is characterized by increased serum NfL levels, which, however, is a biomarker of only motor but not of extra-motor disorders [Verde et al. 2023a]."
This statement seems questionable to me. Please check e.g.,:
JAMA Neurol. doi:10.1001/jamaneurol.2018.3746
http://dx.doi.org/10.1136/jnnp-2018-320106
Some minor flaws still exist.
Author Response
Thank you very much for your help in improving this paper!
"In 11. Conclusions and outlook, the author states: "ALS is characterized by increased serum NfL levels, which, however, is a biomarker of only motor but not of extra-motor disorders [Verde et al. 2023a]."
This statement seems questionable to me. Please check e.g.,:
JAMA Neurol. doi:10.1001/jamaneurol.2018.3746
http://dx.doi.org/10.1136/jnnp-2018-320106"
The statement is now changed and both suggested papers are incorporated. The alteration is now highlighted in blue. Yellow highlighted are still the changes in revision 1.